# Rethinking Evaluation for Temporal Link Prediction through Counterfactual Analysis

## Abstract

In response to critiques of existing evaluation methods for temporal link prediction (TLP) models, we propose a novel approach to verify if these models truly capture temporal patterns in the data. Our method involves a sanity check formulated as a counterfactual question: "What if a TLP model is tested on a temporally distorted version of the data instead of the real data?" Ideally, a TLP model that effectively learns temporal patterns should perform worse on temporally distorted data compared to real data. We provide an in-depth analysis of this hypothesis and introduce two data distortion techniques to assess well-known TLP models. Our contributions are threefold: (1) We introduce two simple techniques to distort temporal patterns within a graph, generating temporally distorted test splits of well-known datasets for sanity checks. These distortion methods are applicable to any temporal graph dataset. (2) We perform counterfactual analysis on six TLP models `JODIE`, `TGAT`, `TGN`, `CAWN`, `GraphMixer`, and `DyGFormer` to evaluate their capability in capturing temporal patterns across different datasets. (3) We introduce two metrics – average time difference (`ATD`) and average count difference (`ACD`) – to provide a comprehensive measure of a model's predictive performance.

## 1 Introduction

In static graphs, link prediction refers to the task of predicting whether an edge exists between two nodes after having observed other edges in the graph. Temporal link prediction (TLP) is a dynamic extension of link prediction wherein the task is to predict whether a link (edge) exists between any two nodes in the future based on the historical observations (Qin and Yeung, 2023). The predictive capability of TLP models make them useful in applications pertaining to dynamic graphs, such as product recommendations (Qin et al., 2024; Fan et al., 2021), social network content or account recommendation (Fan et al., 2019; Daud et al., 2020), fraud detection in financial networks (Kim et al., 2024), and resource allocation, to name a few.

In the TLP literature (Kumar et al., 2019; Trivedi et al., 2019; Xu et al., 2020; Rossi et al., 2020; Wang et al., 2020; Cong et al., 2023; Yu et al., 2023), the TLP task is treated as a binary classification problem where the query

$q_1$ : "Does an edge exist between the nodes $u$ and $v$ at time $t$?"

is processed by a model and then compared with the ground truth following which metrics such as area under the receiver operating characteristic curve (`AU-ROC`), and average precision (`AP`) are reported. The ground truth consists of positive samples, and a fixed number of random negative samples. There are a couple of issues in the binary classification approach. Firstly, the timestamps in the query are restricted to the timestamps present in the ground truth, which makes the evaluation biased and does not test the model's performance in the continuous time range. Secondly, checking for the existence of an edge at a specific timestamp is an ill-posed question, and instead the existence of an edge should be queried within a finite time-interval. Lastly, the negative edge sampling strategy, and the number of negative samples per positive sample impact the performance metrics as seen in `EXH` (Poursafaei and Rabbany, 2023).

Alternatively, in a rank-based approach, the query is formulated as:

$q_2$ : "Which nodes are likely to have an edge with node $u$ at time $t$?"

In this case, the model returns an ordered list of nodes arranged from most likely to least likely. Then, the rank of the ground truth edge is returned if a match is found, and if not, a high number is reported. For all the edges in the test data, metrics such as Mean Average Rank (MAR) or Mean Reciprocal Rank (MRR) can be reported to assess the performance of the model (Huang et al., 2024). While the rank-based metrics are more intuitive than AU-ROC and AP, the issues regarding binary classification mentioned above still remain unaddressed. To give a true picture of the predictive power of the TLP models, a penalty term should be introduced to account for the nodes that are incorrectly estimated to form an edge with node $u$ at time $t$.

In a recent work, Poursafaei et al. (2022) highlighted that the state-of-the-art (SoTA) performance of some TLP models on the standard benchmark datasets is near-perfect. This is counterintuitive because TLP is a challenging task, even more challenging than link prediction of static graphs, due to the additional degree of freedom in the data induced by the temporal dimension. The flaw in the evaluation method is attributed to the limited negative sampling strategy, and the authors propose a new negative edge sampling strategy which results in a different ranking of the baselines.

Inspired by the critique of the evaluation method, we propose a method to conduct sanity check of the TLP models to determine if they truly capture the temporal patterns in the data. The sanity check is formulated as the counterfactual question (Pearl, 2009):

> "What if a TLP model which is trained on a temporal graph is tested on *temporally distorted* version of the data instead of the real data?"

Ideally, a TLP model which is capable of learning the temporal patterns should perform worse on temporally distorted data compared to the real data. We conduct an in-depth analysis of this argument and introduce various data distortion techniques to assess well-known TLP models.

**Contributions**   The contributions of our work can be summarised as follows:

- We introduce simple **techniques** to distort the temporal patterns within a graph. These techniques are then used to generate temporally distorted version of the test split of some famous datasets which can be used for **sanity check**. Moreover, the distortion methods can be applied to any temporal graph dataset [Link to code repository].
- We perform **counterfactual analysis** on TLP models such as JODIE (Kumar et al., 2019), TGAT (Xu et al., 2020), TGN (Rossi et al., 2020), CAWN (Wang et al., 2020), GraphMixer (Cong et al., 2023), and DyGFormer (Yu et al., 2023) to check whether they are capable of capturing the temporal patters within various datasets.
- We propose two **metrics**: average time difference (ATD), and average count difference (ACD) to measure the performance of TLP models. These metrics can provide a holistic picture of a model's predictive performance.
- Lastly, we propose an alternative **evaluation strategy** for TLP through which the existing pitfalls of binary classification and ranking methods can be avoided.

## 2 PRELIMINARIES

### 2.1 DEFINITIONS

In TLP literature, continuous-time temporal graphs with *instantaneous edges* are often considered, where edges represent interaction events between two nodes at a specific point in time. Alternatively, temporal graphs can be defined with edges that appear at a certain time and either persist for a duration (Celikkanat et al., 2024) or accumulate indefinitely. In this work, we focus on the instantaneous edge temporal graph, also known as interaction graphs (Qin et al., 2024) or unevenly sampled edge sequence (Qin and Yeung, 2023).

**Definition 2.1.** A **temporal graph** with $m \in \mathbb{N}$ instantaneous edges formed between nodes in $\mathcal{U}$ and $\mathcal{V}$ is defined as $\mathcal{G} = (\mathcal{U}, \mathcal{V}, \mathcal{E})$, where $\mathcal{E} \triangleq \{(u_i, v_i, t_i) : i \in [m], u_i \in \mathcal{U}, v_i \in \mathcal{V}, t_i \in \mathbb{R}\}$ denotes the set of edges. The tuple $(u, v, t)$ is referred to as an edge event.

While the definition caters to bipartite structure, with $\mathcal{U} = \mathcal{V}$, it can also represent general graphs.

**Definition 2.2.** The occurrences of a particular edge $(u, v)$ in $\mathcal{E}$ is denoted as $\mathcal{E}_{(u,v)}$ and defined as $\mathcal{E}_{(u,v)} \triangleq \{(u, v, t) : (u, v, t) \in \mathcal{E}\}$.

**Definition 2.3.** The slice of edges in $\mathcal{E}$ with timestamps in the range $(t_1, t_2)$ is denoted as $\mathcal{E}(t_1, t_2)$, and defined as $\mathcal{E}(t_1, t_2) \triangleq \{(u, v, t) : (u, v, t) \in \mathcal{E}, t \in (t_1, t_2)\}$.

**Definition 2.4.** The timestamps in $\mathcal{E}$ consisting of $m \in \mathbb{N}$ edges can be extracted through a function $\mathscr{T} : (\mathcal{U} \times \mathcal{V} \times \mathbb{R})^m \to \mathbb{R}^m$ as $\mathscr{T}(\mathcal{E}) \triangleq \{t : (u, v, t) \in \mathcal{E}\}$.

## 2.2 POINT PROCESSES

Perry and Wolfe (2013) modelled the interaction events of a directed edge $(u, v)$ as an inhomogeneous Poisson point process. In a recent work on continuous-time representation learning on temporal graphs, Modell et al. (2024) followed suit, and assumed $\mathcal{E}_{(u,v)}$ to be sampled from an independent inhomogeneous Poisson point process with intensity $\lambda_{(u,v)}(t)$. The number of edge events $(u, v)$ between timestamps $t_1$ and $t_2$ follow a Poisson distribution with rate $\int_{t_1}^{t_2} \lambda_{(u,v)}(t) \, dt$, i.e.,

$$|\mathcal{E}_{(u,v)}(t_1, t_2)| \sim \text{Poisson}\left(\int_{t_1}^{t_2} \lambda_{(u,v)}(t) \, dt\right). \tag{1}$$

To connect the present to the past, Du et al. (2016) view the intensity function $\lambda_{(u,v)}^\star(t)$ as a nonlinear function of the sample history, where $\star$ indicates that the function is conditioned on the history. The conditional density function for edge $(u, v)$ is written as

$$p_{(u,v)}^\star(t) = \lambda_{(u,v)}^\star(t) \exp\left(-\int_{t'}^{t} \lambda_{(u,v)}^\star(\tau) \, d\tau\right), \tag{2}$$

where $t' < t$ is the last time when edge $(u, v)$ was observed. The goal is to find the parameters $\lambda_{(u,v)}^\star(t) : 0 < t \leq T$ which can describe the observation $\mathcal{E}_{(u,v)}$. This is done by minimizing the negative log likelihood (NLL) at the timestamps of edge occurrence (Shchur et al., 2021):

$$\min_{\lambda_{(u,v)}^\star(t) : 0 < t \leq T} - \sum_{t \in \mathscr{T}\left(\mathcal{E}_{(u,v)}\right)} \log\left(\lambda_{(u,v)}^\star(t)\right) + \int_0^T \lambda_{(u,v)}^\star(\tau) \, d\tau, \quad T = \max \mathscr{T}\left(\mathcal{E}_{(u,v)}\right). \tag{3}$$

Shchur et al. (2021) summarize the operation of a neural temporal point process as follows:

- The edge events in $\{(u, v, t_i) : i \in [m]\}$ are represented as feature vectors $\boldsymbol{x}_i = f_{\mathfrak{e}}(u, v, t_i)$,
- The historical feature vectors are encoded into a state vector $\boldsymbol{h}_i = f_{\mathfrak{h}}(\boldsymbol{x}_1, \cdots \boldsymbol{x}_{i-1})$,
- The distribution of $t_i$ conditioned on the past is simply conditioned on $\boldsymbol{h}_i$.

The functions $f_{\mathfrak{e}}$ and $f_{\mathfrak{h}}$, as well as the conditioning on $\boldsymbol{h}_i$ are implemented using neural networks.

## 3 COUNTERFACTUAL ANALYSIS

A temporal graph is characterized by (1) the *order* in which the edges appear, (2) the *frequency* with which edges appear over time, and (3) the *time gap* between any two edge events. In this work, we refer to these characteristics as **temporal patterns**. Furthermore, if temporal patterns observed in the past enable predictions of future temporal patterns that outperform naïve estimates on a specific performance metric, then the temporal data is considered **learnable**. This does not require the temporal pattern to remain consistent over time; rather, it suggests that future changes can be estimated from past observations.

**Experiment Setup** A model $f$ is trained on a temporal graph $\mathcal{E}_{\text{train}}$ and tested on $\mathcal{E}_{\text{test}}$ through the binary classification approach resulting in a performance metric such as AP. The train and test data are chronologically split from the same temporal graph which is assumed to be generated through a common causal mechanism, i.e., $\mathcal{E}_{\text{train}} = \mathcal{E}(0, \tau_0)$, and $\mathcal{E}_{\text{test}} = \mathcal{E}(\tau_0, T)$. In light of the experimental setup, we ask the following question:

> Would the model $f$ which is trained on $\mathcal{E}_{\text{train}}$ perform well if tested on a distorted version of $\mathcal{E}_{\text{test}}$ instead of $\mathcal{E}_{\text{test}}$?

To formalise the question in the counterfactual framework (Pearl, 2019), we consider the following statements:

$x'$ : The model $f$ is tested on $\mathcal{E}_{\text{test}}$
$y'$: The performance metric is $\alpha$
$x$ : The model $f$ is tested on a *temporally distorted* version of $\mathcal{E}_{\text{test}}$
$y$ : The performance metric is less than $\alpha$

Additionally, $y_x$ is read as $y$ when $x$. The counterfactual question is framed as $P\left(y_x \mid x', y'\right)$, i.e.,

The probability that the performance metric would be less than $\alpha$ had the test data been a temporally distorted version of $\mathcal{E}_{\text{test}}$, given the performance metric was observed to be at least $\alpha$ when the model was tested on $\mathcal{E}_{\text{test}}$.

To answer the question above, we design the *intervention* as graphically depicted in Fig. 1. The TLP model $f$ is trained on the data $\mathcal{E}_{\text{train}}$. The true test data $\mathcal{E}_{\text{test}}$ is temporally distorted through some function $\mathfrak{D}(\cdot)$ resulting in $\mathcal{E}' = \mathfrak{D}(\mathcal{E}_{\text{test}})$. Finally, we test the model $f$ on the true data $\mathcal{E}_{\text{test}}$ and the temporally distorted data $\mathcal{E}'$ and compare the metrics which may result in either of the two scenarios shown in the figure based on which we can comment on the effectiveness of $f$.

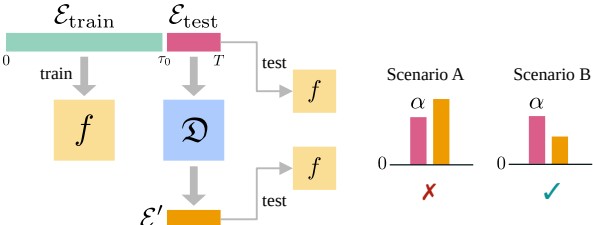

Figure 1: The intervention setup to verify the counterfactual question above.

**Motivation** To motivate the counterfactual analysis, we present a simplified example of *classification on binary sequences*, viewed as a discretized version of the temporal graph described in Def. 2.1. Let the set of all binary sequences of length $m \in \mathbb{N}$ be denoted by $\mathbb{B}_m = \{0, 1\}^m$, and let $\boldsymbol{b} \in \mathbb{B}_m$ be a binary sequence representing the true data. Moreover, we consider a model $f$ whose output is $\hat{b} \in \mathbb{B}_m$. The performance metric achieved by $\hat{b}$ on ground truth $\boldsymbol{b}$ is denoted as $\phi(\hat{b}, \boldsymbol{b})$. Next, let $\boldsymbol{b}' \in \mathbb{B}_m \setminus \{\boldsymbol{b}\}$ denote a distorted version of $\boldsymbol{b}$. Building on the above setup, the counterfactual question is $P(\phi(\hat{b}, \boldsymbol{b}') < \phi(\hat{b}, \boldsymbol{b}))$, i.e., the probability that the model performs relatively worse on the distorted sequence. Next, we find the **conditions** on model output $\hat{b}$ and distorted sequence $\boldsymbol{b}'$ such that $P(\phi(\hat{b}, \boldsymbol{b}') < \phi(\hat{b}, \boldsymbol{b})) = 1$.

Figure 2: Scatter plot showing the normalised Hamming distance between $\boldsymbol{b}$ and $\boldsymbol{b}'$ on the x-axis and the performance (AP) of the classifier on the distorted sequence $\boldsymbol{b}'$ on the y-axis. The normalised Hamming distance serves as a distortion metric for binary sequences. Each point corresponds to a random $\boldsymbol{b} \in \mathbb{B}_m$, and $\hat{b}$ such that $\phi(\hat{b}, \boldsymbol{b}) \geq \alpha$. In the figure, $m = 16$, and $\alpha = 0.9$. We observe that for $\|\boldsymbol{b} - \boldsymbol{b}'\|_1 > \beta$, all the points lie below $\alpha$, i.e., $P(\phi(\hat{b}, \boldsymbol{b}') < \alpha) = 1$.

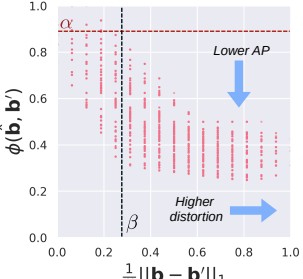

Let $\mathfrak{M}$ be a **causal model** which generates $\boldsymbol{b}$ succeeding another binary sequence $\boldsymbol{b}_0$. The causal model can produce multiple sequences succeeding $\boldsymbol{b}_0$, i.e., $\boldsymbol{b} \sim \mathfrak{M}(\boldsymbol{b}_0)$. We assume that $\mathfrak{M}(\boldsymbol{b}_0) \in \Omega \subset \mathbb{B}_m$ with $|\Omega| \ll 2^m$. If the model $f$ produces output $\hat{b} \in \Omega$ after being trained on $\boldsymbol{b}_0$, we can say that it has learnt the causal mechanism underlying $\mathfrak{M}$, which gives us the lower bound on the performance metric as $\alpha_0 = \inf_{\hat{b}, \boldsymbol{b} \in \Omega} \phi(\hat{b}, \boldsymbol{b})$. Then, based on Fig. 2, we make the following Assumption:

**Assumption 3.1.** For some $\alpha \leq \alpha_0$, there exists $\beta \in (0, 1)$ such that a model which admits a performance metric of $\alpha$ on the true sequence $\boldsymbol{b}$, reports a performance metric lower than $\alpha$ for all distorted samples $\boldsymbol{b}'$ with distortion greater than $\beta$.

Going back to the experiment setup in Fig. 1, we conjecture the following:

**Conjecture 3.1.** *$P(y_x \mid x', y') \neq 1 \implies$ model $f$ is not capable of discerning the temporal patterns distorted through $\mathfrak{D}$.*

We now present the logic behind this conjecture. Please consider the following statements,

$s_1$ : The model $f$ is capable of discerning temporal patterns in $(\mathcal{E}_{\text{train}}, \mathcal{E}_{\text{test}})$
$s_2$ : The function $\mathfrak{D}$ generates temporally distorted test data $\mathcal{E}' = \mathfrak{D}(\mathcal{E}_{\text{test}})$
$s_3$ : The data $(\mathcal{E}_{\text{train}}, \mathcal{E}_{\text{test}})$ is learnable
$s_4$ : The performance metric reported by the model $f$ on true test data $\mathcal{E}_{\text{test}}$ is always higher than that reported on the distorted test data $\mathcal{E}'$, i.e., $P(y_x \mid x', y') = 1$.

We start with $s_1 \wedge s_2 \wedge s_3 \implies s_4$. Assuming that the data is learnable, i.e., $s_3 = 1$, we get $s_1 \wedge s_2 \implies s_4$. Through contraposition, we arrive at $\neg s_4 \implies \neg s_1 \vee \neg s_2$, where

$$\neg s_4 \equiv P(y_x \mid x', y') \neq 1.$$

Further, we impose that $\mathfrak{D}$ satisfies Assumption 3.1, i.e., $\neg s_2 = 0$, allowing us to conclude $\neg s_4 \implies \neg s_1$ which reads as the conjecture above.

In Fig. 3 we present an example depicting what temporal distortion means using point processes. In the remainder of this section, we define *temporal distortion metrics* and then discuss two *temporal distortion techniques* to distort the temporal graphs.

Figure 3: Let $\mathcal{E}_{\text{train}} \cup \mathcal{E}_{\text{test}}$ be sampled from a point process with intensity $\lambda^\star(t), t \in [0, T]$. We generate $\mathcal{E}'$ from another point process with intensity $\lambda'(t), t \in [\tau_0, T]$. We depict the intensity functions as two sinusoidal waves with different frequency and phase. If a model $f$ learns this intensity function by observing $\mathcal{E}_{\text{train}}$, and then generates samples for prediction, they would be more similar to $\mathcal{E}_{\text{test}}$ than $\mathcal{E}'$, resulting in a lower performance on the distorted test set.

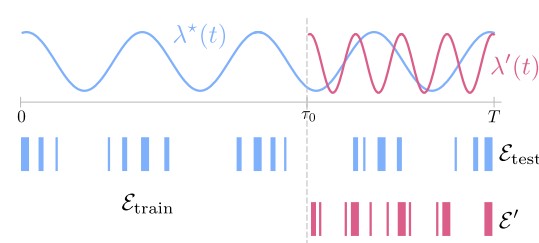

### 3.1 TEMPORAL DISTORTION METRICS

Let $\mathcal{E}$ be a temporal graph sampled from a temporal point process with intensity $\lambda^\star(t)$ for $t \in [0, T]$. Let $\mathcal{E}'$ be data sampled from another point process with intensity $\lambda'(t)$ for $t \in [0, T]$.

**Definition 3.1.** The temporal graph $\mathcal{E}'$ is $\delta$-temporally distorted w.r.t. $\mathcal{E}$ if for some $\delta > 0$,

$$\sum_{(u,v) \in \mathcal{U} \times \mathcal{V}} \frac{1}{T} \int_0^T |\lambda^*_{(u,v)}(t) - \lambda'_{(u,v)}(t)| \, dt > \delta. \tag{4}$$

In practice, we do not have access to the true intensity functions, so we have to compare the realisations instead. Let $\mathcal{E}$ and $\mathcal{E}'$ be two temporal graphs, then we measure the difference in their temporal patterns through the metrics defined below.

**Definition 3.2.** The average time difference (ATD) between $\mathcal{E}$ and $\mathcal{E}'$ is defined as:

$$\mathsf{ATD}(\mathcal{E}, \mathcal{E}') \triangleq \frac{1}{T|\mathcal{E}|} \sum_{(u,v,t) \in \mathcal{E}} \min_{t' \in \mathscr{T}(\mathcal{E}'_{(u,v)}) \cup \{T\}} |t - t'|, \quad T = \max \mathscr{T}(\mathcal{E}) - \min \mathscr{T}(\mathcal{E}). \tag{5}$$

In ATD, we measure the time difference between the edge event $(u, v, t) \in \mathcal{E}$ and the closest $(u, v, t') \in \mathcal{E}'$, reporting the average over all the edge events in $\mathcal{E}$. In Fig. 4 we show two temporal graphs as *impulse trains* with each impulse color coded to represent the edge of the sample 3-node graph. Through ATD we can measure the overall difference in the occurrence of an edge event. However, ATD fails to capture the difference in the frequency with which an edge occurs in the two temporal graphs $\mathcal{E}$ and $\mathcal{E}'$. Therefore, we define average count difference ACD to measure the difference in the frequency with which edges occur in the temporal graph.

**Definition 3.3.** The average count difference (ACD) between $\mathcal{E}$ and $\mathcal{E}'$ is defined as:

$$\mathsf{ACD}(\mathcal{E}, \mathcal{E}') \triangleq \frac{1}{|\mathcal{E}|} \sum_{(u,v,t) \in \mathcal{E}} \left| |\mathcal{E}_{(u,v)}(t - \bar{\tau}, t + \bar{\tau})| - |\mathcal{E}'_{(u,v)}(t - \bar{\tau}, t + \bar{\tau})| \right|, \quad \bar{\tau} \in \mathbb{R}^+. \quad (6)$$

For each edge event $(u, v, t) \in \mathcal{E}$, we count the number of occurrences of $(u, v)$ in the time range $(t - \bar{\tau}, t + \bar{\tau})$ in both $\mathcal{E}$ and $\mathcal{E}'$ and measure the count difference. In Fig. 4 we depict the time interval as a light blue box centred around each edge event in $\mathcal{E}$. For $\bar{\tau} \to 0$, the search becomes restricted to an infinitesimal time interval, with $\mathsf{ACD}(\mathcal{E}, \mathcal{E}')$ approaching $1 - \frac{1}{|\mathcal{E}|} \sum_{(u,v,t) \in \mathcal{E}} \mathbb{I}\{(u, v, t) \in \mathcal{E}'\}$.

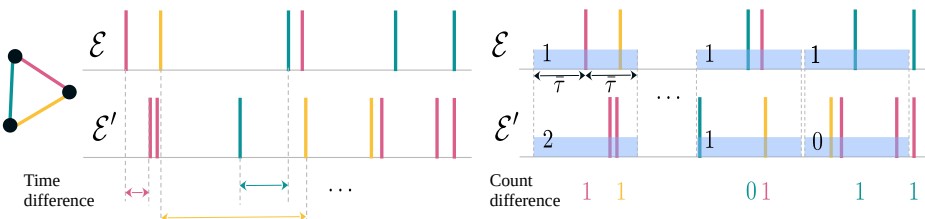

Figure 4: Comparing two temporal graphs by measuring the time difference, and the count difference within intervals of duration $2\bar{\tau}$ centred around the edge event.

Table 1: Joint interpretation of the distortion metrics.

|  | ATD $\downarrow$ | ATD $\uparrow$ |
| --- | --- | --- |
| ACD $\downarrow$ | similar | the edge events are shifted near the extremities of the $2\bar{\tau}$ interval. |
| ACD $\uparrow$ | the edge event is duplicated multiple times in the vicinity of the original edge event. | the edge events are either duplicated or reduced, and also shifted away from the original edge interval. |

## 3.2 TEMPORAL DISTORTION TECHNIQUES

Now that we are equipped with metrics to measure the difference between two temporal graphs, we device distortion functions $\mathfrak{D}(\cdot)$ which can enable us to investigate the counterfactual question posed earlier. We propose two distortion techniques $\mathfrak{D}_{\text{INTENSE}}(\cdot, K)$ which creates $K$ time-perturbed copies of each edge events, and $\mathfrak{D}_{\text{SHUFFLE}}(\cdot)$ wherein the timestamps of different edge events are shuffled.

**INTENSE** Let the real temporal graph data be denoted by $\mathcal{E} = \cup_{(u,v) \in \mathcal{U} \times \mathcal{V}} \mathcal{E}_{(u,v)}$, and the distorted version be denoted by $\mathcal{E}' = \cup_{(u,v) \in \mathcal{U} \times \mathcal{V}} \mathcal{E}'_{(u,v)}$. Then, for each edge event $(u, v, t)$ in the real data $\mathcal{E}$, we create $K$ edge events $(u, v, t + \tau)$ with $\tau$ sampled uniformly from $(-\bar{\tau}, \bar{\tau})$ for some $\bar{\tau} \in \mathbb{R}^{+1}$. Alternatively, if it is known that $\mathcal{E}_{(u,v)}$ is sampled from a point process with intensity $\lambda^\star_{(u,v)}(t)$, then we can generate $\mathcal{E}'_{(u,v)}$ by sampling from another point process with intensity $\lambda'_{(u,v)}(t)$, such that

$$\lambda'_{(u,v)}(t) = K\lambda^\star_{(u,v)}(t), \quad \forall (u, v) \in \mathcal{U} \times \mathcal{V}. \quad (7)$$

**SHUFFLE** For any two edge events $(u, v, t), (u', v', t') \in \mathcal{E}$, we shuffle the timestamps in the distorted version, i.e. $(u, v, t'), (u', v', t) \in \mathcal{E}'$. The shuffling process is also called label permutation (Chatterjee, 2018). In terms of the point process, we can explain shuffling as follows. If $\mathcal{E}_{(u,v)}$ is known to be sampled from a point process with intensity $\lambda^\star_{(u,v)}(t)$, then $\mathcal{E}'_{(u,v)}$ can be generated by sampling from an inhomogeneous Poisson point process with intensity $\lambda'_{(u,v)}(t)$, where

$$\lambda'_{(u,v)}(t) = \frac{\left(\int_0^T \lambda^\star_{(u,v)}(t)\,dt\right) \sum_{(u',v') \in \mathcal{U} \times \mathcal{V}} \lambda^\star_{(u',v')}(t)}{\sum_{(u',v') \in \mathcal{U} \times \mathcal{V}} \int_0^T \lambda^\star_{(u',v')}(t)\,dt}. \quad (8)$$

**Algorithm 1** $\mathfrak{D}_{\text{INTENSE}}$

**Input** $\mathcal{E}, K \in \mathbb{N}, \bar{\tau} \in \mathbb{R}^+$
**Output** $\mathcal{E}'$
1:  $\mathcal{E}' = \varnothing$
2: **for** $(u, v, t) \in \mathcal{E}$ **do**
3:    **for** $k \in [K]$ **do**
4:      $\tau \sim \text{Uniform}(-\bar{\tau}, \bar{\tau})$
5:      $\mathcal{E}' \leftarrow \mathcal{E}' \cup \{(u, v, t + \tau)\}$
6:    **end for**
7: **end for**

**Algorithm 2** $\mathfrak{D}_{\text{SHUFFLE}}$

**Input** $\mathcal{E}$
**Output** $\mathcal{E}'$
1:  $\mathcal{E}' = \varnothing$
2: $\mathcal{T} \leftarrow \mathscr{T}(\mathcal{E})$
3: **for** $(u, v, t) \in \mathcal{E}$ **do**
4:    $\tau \sim \mathcal{T}$
5:    $\mathcal{E}' \leftarrow \mathcal{E}' \cup \{(u, v, \tau)\}$
6:    $\mathcal{T} \leftarrow \mathcal{T} \setminus \{\tau\}$
7: **end for**

The operations of $\mathfrak{D}_{\text{INTENSE}}$ and $\mathfrak{D}_{\text{SHUFFLE}}$ are described in Algorithms 1 and 2, respectively. Moreover, a visual example is provided in Fig. 5. The computational complexity of $\mathfrak{D}_{\text{INTENSE}}(\cdot, K)$ is $\mathcal{O}(K|\mathcal{E}|)$, and of $\mathfrak{D}_{\text{SHUFFLE}}(\cdot)$ is $\mathcal{O}(|\mathcal{E}|)$. In short, both distortion techniques are linear in the number of edge events in $\mathcal{E}$.

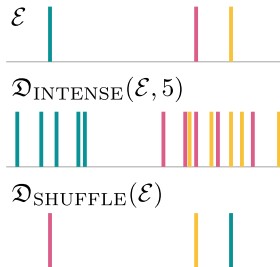

Figure 5: Visual representation of INTENSE and SHUFFLE distortions. In $\mathfrak{D}_{\text{INTENSE}}(\mathcal{E}, 5)$, 5 edge events are created in the vicinity of the true edge event in $\mathcal{E}$. This increases the frequency with which edges appear in an interval, thereby distorting the temporal pattern. In $\mathfrak{D}_{\text{SHUFFLE}}(\mathcal{E})$, as the name suggests, the order in which the edges appear is shuffled and thus the temporal pattern is distorted, the edges now appear where they should not be.

We use $\mathfrak{D}_{\text{INTENSE}}(\cdot, 5)$ and $\mathfrak{D}_{\text{SHUFFLE}}(\cdot)$ to create 10 temporally distorted samples of the test splits of each dataset. In Table 2, we present the ATD, and ACD by comparing the distorted samples with the original test data of different datasets. The metrics ATD and ACD should be considered in conjunction to measure the dissimilarity of two temporal graphs. For each real test data, we create 10 distorted samples and report the mean and 95% confidence interval of the metrics to ensure statistical reliability.

Table 2: Distortion measures on different datasets.

|  |  | wikipedia | reddit | uci | lastfm | mooc |
|---|---|---|---|---|---|---|
| INTENSE | ATD | 6.9e-6 $\pm$ 2e-8 | 1.6e-6 $\pm$ 2e-9 | 1.6e-5 $\pm$ 1.2e-7 | 8.6e-7 $\pm$ 9.4e-10 | 2.5e-6 $\pm$ 7e-9 |
|  | ACD | 4.479 $\pm$ 1.9e-3 | 4.112 $\pm$ 3.9e-4 | 7.214 $\pm$ 1.2e-2 | 4.046 $\pm$ 1.8e-4 | 4.627 $\pm$ 1.4e-3 |
| SHUFFLE | ATD | 0.078 $\pm$ 5.7e-4 | 0.099 $\pm$ 3e-4 | 0.132 $\pm$ 8.4e-4 | 0.0800 $\pm$ 1.7e-4 | 0.1906 $\pm$ 6.7e-4 |
|  | ACD | 1.093 $\pm$ 3.4e-4 | 1.033 $\pm$ 8e-5 | 1.877 $\pm$ 3.3e-3 | 1.0011 $\pm$ 1.4e-4 | 1.1896 $\pm$ 8.9e-5 |

## 4 RESULTS

We evaluate the performance of the following TLP models in light of Proposition 3.1: JODIE (Kumar et al., 2019), TGAT (Xu et al., 2020), TGN (Rossi et al., 2020), CAWN (Wang et al., 2020), GraphMixer (Cong et al., 2023), DyGFormer (Yu et al., 2023)

The models are evaluated under two **settings**: *transductive* and *inductive*. In transductive TLP, the nodes $u, v$ in the positive sample $(u, v, t) \in \mathcal{E}_{\text{test}}$ were observed during training. In contrast, in inductive TLP, at least one node in $u, v$ is novel, and was not observed during training.

In Table 3, we have arranged the datasets in increasing order of their size (more details can be found in Appendix A.1). We notice that all the TLP models pass the counterfactual test for SHUFFLE distortion on the smallest dataset: uci, and some of them {TGAT, GraphMixer, DyGFormer}

---
[1]For the experiments we set $\bar{\tau} = \frac{\max \mathscr{T}(\mathcal{E}) - \min \mathscr{T}(\mathcal{E})}{|\mathcal{E}|}$.

pass for SHUFFLE on the second-smallest dataset `wikipedia`, and only `GraphMixer` and `TGN` pass on `reddit`. Surprisingly, `JODIE` passes on INTENSE distortion for two of the largest datasets `lasstfm` and `mooc`. And overall, none of the TLP models pass the counterfactual test on the INTENSE distortions. This allows us to conclude the followg: (1) The TLP models are able to discern the temporarl order of edge occurrence, however this capability worsens for larger datasets, and (2) the TLP models do not keep count of the frequency with which edges appear over time.

Table 3: Performance (AP) of the models `JODIE`, `TGAT`, `TGN`, `CAWN`, `GraphMixer`, and `DyGFormer` on five datasets, and their temporally distorted versions denoted as INTENSE, and SHUFFLE. For each metric, we report the mean, and the 95% confidence interval (CI) as mean ± CI. We have marked the metrics in blue for distortions that showed that a model was incapable of learning on a certain dataset as per Conjecture 3.1, and orange otherwise.

| | AP | uci | wikipedia | reddit | lastfm | mooc |
|---|---|---|---|---|---|---|
| **JODIE** | *transductive* | $0.8726 \pm 5e\text{-}3$ | $0.9137 \pm 5e\text{-}3$ | $0.9654 \pm 5e\text{-}3$ | $0.7036 \pm 2e\text{-}3$ | $0.8068 \pm 6e\text{-}4$ |
| | INTENSE | $0.9129 \pm 5e\text{-}3$ | $0.9078 \pm 1e\text{-}2$ | $0.9567 \pm 1e\text{-}2$ | $0.7090 \pm 3e\text{-}4$ | $0.7556 \pm 4e\text{-}4$ |
| | SHUFFLE | $0.8509 \pm 3e\text{-}3$ | $0.8962 \pm 4e\text{-}2$ | $0.9613 \pm 4e\text{-}2$ | $0.7036 \pm 1e\text{-}3$ | $0.8072 \pm 5e\text{-}4$ |
| | *inductive* | $0.7310 \pm 2e\text{-}2$ | $0.8970 \pm 5e\text{-}3$ | $0.9138 \pm 2e\text{-}2$ | $0.8431 \pm 4e\text{-}3$ | $0.7931 \pm 1e\text{-}3$ |
| | INTENSE | $0.8332 \pm 8e\text{-}3$ | $0.8972 \pm 1e\text{-}2$ | $0.9308 \pm 4e\text{-}2$ | $0.8361 \pm 5e\text{-}4$ | $0.7658 \pm 3e\text{-}4$ |
| | SHUFFLE | $0.6994 \pm 8e\text{-}3$ | $0.9078 \pm 2e\text{-}2$ | $0.9251 \pm 6e\text{-}3$ | $0.8431 \pm 2e\text{-}3$ | $0.7931 \pm 7e\text{-}4$ |
| **TGAT** | *transductive* | $0.7694 \pm 7e\text{-}3$ | $0.9528 \pm 2e\text{-}3$ | $0.9818 \pm 6e\text{-}4$ | $0.7309 \pm 3e\text{-}4$ | $0.8458 \pm 5e\text{-}4$ |
| | INTENSE | $0.8637 \pm 2e\text{-}2$ | $0.9691 \pm 2e\text{-}3$ | $0.9825 \pm 6e\text{-}4$ | $0.9840 \pm 1e\text{-}4$ | $0.9610 \pm 1e\text{-}4$ |
| | SHUFFLE | $0.7336 \pm 2e\text{-}2$ | $0.9532 \pm 5e\text{-}3$ | $0.9826 \pm 6e\text{-}3$ | $0.7308 \pm 3e\text{-}4$ | $0.8458 \pm 4e\text{-}4$ |
| | *inductive* | $0.7008 \pm 1e\text{-}2$ | $0.9401 \pm 2e\text{-}3$ | $0.9658 \pm 1e\text{-}3$ | $0.7817 \pm 2e\text{-}4$ | $0.8430 \pm 3e\text{-}4$ |
| | INTENSE | $0.8095 \pm 2e\text{-}2$ | $0.9621 \pm 2e\text{-}3$ | $0.9676 \pm 1e\text{-}3$ | $0.9841 \pm 1e\text{-}4$ | $0.9621 \pm 1e\text{-}4$ |
| | SHUFFLE | $0.6324 \pm 1e\text{-}2$ | $0.9304 \pm 7e\text{-}3$ | $0.9664 \pm 3e\text{-}3$ | $0.7817 \pm 2e\text{-}4$ | $0.8430 \pm 3e\text{-}4$ |
| **TGN** | *transductive* | $0.7975 \pm 1e\text{-}2$ | $0.9472 \pm 1e\text{-}3$ | $0.9578 \pm 1e\text{-}3$ | $0.7764 \pm 5e\text{-}3$ | $0.8855 \pm 4e\text{-}3$ |
| | INTENSE | $0.9709 \pm 3e\text{-}3$ | $0.9911 \pm 6e\text{-}4$ | $0.9744 \pm 2e\text{-}3$ | $0.9916 \pm 1e\text{-}5$ | $0.9629 \pm 6e\text{-}4$ |
| | SHUFFLE | $0.6520 \pm 4e\text{-}2$ | $0.8487 \pm 3e\text{-}2$ | $0.9563 \pm 2e\text{-}3$ | $0.7764 \pm 9e\text{-}4$ | $0.8848 \pm 1e\text{-}3$ |
| | *inductive* | $0.7948 \pm 6e\text{-}3$ | $0.9463 \pm 1e\text{-}3$ | $0.9346 \pm 1e\text{-}3$ | $0.8336 \pm 4e\text{-}3$ | $0.8873 \pm 1e\text{-}3$ |
| | INTENSE | $0.9650 \pm 2e\text{-}3$ | $0.9908 \pm 6e\text{-}4$ | $0.9645 \pm 3e\text{-}3$ | $0.9927 \pm 2e\text{-}5$ | $0.9641 \pm 2e\text{-}4$ |
| | SHUFFLE | $0.6193 \pm 9e\text{-}3$ | $0.8376 \pm 3e\text{-}2$ | $0.9299 \pm 3e\text{-}3$ | $0.8337 \pm 6e\text{-}3$ | $0.8872 \pm 2e\text{-}3$ |
| **CAWN** | *transductive* | $0.9397 \pm 8e\text{-}4$ | $0.9901 \pm 1e\text{-}4$ | $0.9884 \pm 3e\text{-}3$ | $0.8755 \pm 3e\text{-}4$ | $0.8667 \pm 2e\text{-}4$ |
| | INTENSE | $0.9889 \pm 7e\text{-}4$ | $0.9975 \pm 8e\text{-}5$ | $0.9942 \pm 7e\text{-}5$ | $0.9879 \pm 2e\text{-}4$ | $0.9719 \pm 1e\text{-}4$ |
| | SHUFFLE | $0.8866 \pm 2e\text{-}3$ | $0.9887 \pm 3e\text{-}4$ | $0.9880 \pm 2e\text{-}3$ | $0.8755 \pm 3e\text{-}4$ | $0.8666 \pm 3e\text{-}4$ |
| | *inductive* | $0.9273 \pm 2e\text{-}3$ | $0.9896 \pm 4e\text{-}4$ | $0.9859 \pm 3e\text{-}3$ | $0.9031 \pm 5e\text{-}4$ | $0.8543 \pm 4e\text{-}4$ |
| | INTENSE | $0.9857 \pm 2e\text{-}3$ | $0.9971 \pm 1e\text{-}5$ | $0.9938 \pm 8e\text{-}5$ | $0.9889 \pm 3e\text{-}4$ | $0.9731 \pm 2e\text{-}4$ |
| | SHUFFLE | $0.8783 \pm 3e\text{-}2$ | $0.9896 \pm 6e\text{-}3$ | $0.9851 \pm 1e\text{-}3$ | $0.9030 \pm 5e\text{-}4$ | $0.8541 \pm 4e\text{-}4$ |
| **GraphMixer** | *transductive* | $0.9323 \pm 2e\text{-}3$ | $0.9690 \pm 4e\text{-}4$ | $0.9738 \pm 3e\text{-}4$ | $0.7630 \pm 1e\text{-}4$ | $0.8233 \pm 3e\text{-}4$ |
| | INTENSE | $0.9923 \pm 6e\text{-}4$ | $0.9966 \pm 2e\text{-}4$ | $0.9965 \pm 1e\text{-}4$ | $0.9858 \pm 1e\text{-}4$ | $0.9537 \pm 1e\text{-}4$ |
| | SHUFFLE | $0.8553 \pm 3e\text{-}3$ | $0.9096 \pm 1e\text{-}3$ | $0.9725 \pm 2e\text{-}4$ | $0.7630 \pm 1e\text{-}4$ | $0.8230 \pm 2e\text{-}4$ |
| | *inductive* | $0.9133 \pm 1e\text{-}3$ | $0.9639 \pm 1e\text{-}4$ | $0.9517 \pm 8e\text{-}4$ | $0.8261 \pm 3e\text{-}4$ | $0.8077 \pm 2e\text{-}4$ |
| | INTENSE | $0.9771 \pm 5e\text{-}4$ | $0.9939 \pm 1e\text{-}4$ | $0.9937 \pm 2e\text{-}4$ | $0.9867 \pm 1e\text{-}4$ | $0.9555 \pm 1e\text{-}4$ |
| | SHUFFLE | $0.7945 \pm 3e\text{-}4$ | $0.8900 \pm 2e\text{-}3$ | $0.9477 \pm 7e\text{-}4$ | $0.8261 \pm 3e\text{-}4$ | $0.8072 \pm 3e\text{-}4$ |
| **DyGFormer** | *transductive* | $0.9596 \pm 3e\text{-}4$ | $0.9901 \pm 2e\text{-}4$ | $0.9921 \pm 1e\text{-}4$ | $0.9096 \pm 1e\text{-}4$ | $0.8622 \pm 2e\text{-}4$ |
| | INTENSE | $0.9938 \pm 1e\text{-}4$ | $0.9983 \pm 1e\text{-}4$ | $0.9984 \pm 1e\text{-}4$ | $0.9912 \pm 1e\text{-}4$ | $0.9709 \pm 1e\text{-}4$ |
| | SHUFFLE | $0.9515 \pm 1e\text{-}3$ | $0.9892 \pm 1e\text{-}4$ | $0.9924 \pm 1e\text{-}4$ | $0.9096 \pm 2e\text{-}4$ | $0.8620 \pm 4e\text{-}4$ |
| | *inductive* | $0.9437 \pm 1e\text{-}4$ | $0.9854 \pm 5e\text{-}4$ | $0.9880 \pm 3e\text{-}4$ | $0.9293 \pm 1e\text{-}4$ | $0.8509 \pm 3e\text{-}4$ |
| | INTENSE | $0.9854 \pm 1e\text{-}4$ | $0.9965 \pm 4e\text{-}5$ | $0.9973 \pm 1e\text{-}5$ | $0.9918 \pm 2e\text{-}4$ | $0.9723 \pm 1e\text{-}4$ |
| | SHUFFLE | $0.9291 \pm 4e\text{-}4$ | $0.9833 \pm 3e\text{-}4$ | $0.9878 \pm 3e\text{-}4$ | $0.9293 \pm 1e\text{-}4$ | $0.8506 \pm 5e\text{-}4$ |

## 5 DISCUSSION

Some of the TLP models used in this work such as `GraphMixer`, and `DyGFormer` are considered the SoTA on most datasets, with near-perfect performance. However, as we showed earlier, a higher metric alone is not indicative of good performance without sanity checks. The counterfactual question helps make the evaluation more explainable, as models that perform worse on temporally distorted data with high ATD and ACD can claim superiority over models that do not. An ideal TLP model should be able to capture the difference in the count of edge events, their order, and the temporal shifts in the edge events.

To reiterate, if the performance of the model on the temporally distorted test data is similar or better than the performance on the original test data, then it implies one the following:

- the model has not made use of the temporal information in the training set,
- there is no useful temporal information in the dataset,
- the temporal distortion is weak.

In the absence of a guarantee that the dataset has useful temporal information that can aid prediction, we can compare different models by comparing the performance gaps.

**Future Work**   Moving away from the binary classification approach to assess the performance of temporal link prediction, future research should explore a generative approach where after observing a temporal graph from time $t \in (0, \tau_0)$, the model can generate a temporal graph in $t \in (\tau_0, T)$. This generated temporal graph can then be compared with the ground truth to measure similarity and assess the performance of the model. The proposed metrics ATD and ACD can be used to measure the difference in the timestamps, as well as the occurrence frequency of the edges. Furthermore, the same model architecture used by the TLP models discussed in this paper, can be further improved by devising new training objectives that incorporate counterfactual analysis through the distortions SHUFFLE and INTENSE.

**Conclusion**   In this work, rather than introducing novel datasets, we present techniques for generating temporally distorted versions of any temporal graph dataset. This makes the contribution relevant even for datasets which will be introduced in the future. To the best of our knowledge, we are the first to apply counterfactual analysis to TLP and hope that it can help standardize the assessment of TLP models.

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

## A  DATASETS & MODELS

### A.1  TEMPORAL GRAPH DATASETS

We use the following datasets[2] to perform counterfactual analysis[3]:

- `wikipedia` (Kumar et al., 2019) describes a dynamic graph of interaction between the editors and Wikipedia pages over a span of one month. The entries consist of the user ID, page ID, and timestamp. The edge features are LIWC-feature vectors (Pennebaker et al., 2001) of the edit text. The edge feature dimension is 172.

- `reddit` (Kumar et al., 2019) describes a bipartite interaction graph between the users and subreddits. The interaction event is recorded with the IDs of the user, subreddit and timestamp. Similar to `wikipedia`, the post content is converted into a LIWC-feature vector of dimension 172 which serves as the edge feature.

- `uci` (Panzarasa et al., 2009) is a dynamic graph describing message-exchange among the students at University of California at Irvine (UCI) from April to October 2004. The interaction event consists of the user IDs, and timestamp.

- `lastfm` (Kumar et al., 2019) is also a bipartite graph depicting the interactions between 1000 users and 1000 most listened songs over a span of one month.

- `mooc` (Kumar et al., 2019) as the name suggests is a student interaction network enrolled in the same online course.

Table 4: The scale of different datasets.

| Dataset | Total nodes ($10^3$) | Total Edges ($10^3$) | Unique Edges ($10^3$) |
|---|---|---|---|
| uci | 1.89 | 59.84 | 20.29 |
| wikipedia | 9.23 | 157.47 | 18.25 |
| reddit | 10.98 | 672.45 | 78.52 |
| lastfm | 1.98 | 1293.10 | 154.99 |
| mooc | 7.14 | 411.75 | 178.44 |

### A.2  TEMPORAL LINK PREDICTION MODELS

We make use of the following models[4] to test the counterfactual framework:

- `JODIE` (Kumar et al., 2019) uses a recurrent neural network (RNN) to generate node embeddings for each interaction event. The future embedding of a node is estimated through a novel projection operator which is turn in used to predict future edge events.

- `TGAT` (Xu et al., 2020) relies on self-attention mechanism to generate node embeddings to capture the temporal evolution of the graph structure.

- `TGN` (Rossi et al., 2020) combine memory modules with graph-based operators to create an encoder-decoder pair capable of creating temporal node embeddings.

- `CAWN` (Wang et al., 2020) propose a novel strategy based on the law of triadic closure, where temporal walks retrieve the dynamic graph motifs without explicitly counting and selecting the motifs. The node IDs are replaced with the hitting counts to facilitate inductive inference.

- `GraphMixer` (Cong et al., 2023) use a simple architecture where the encoder and decoder are designed using multi-layer perceptrons (MLPs).

- `DyGFormer` (Yu et al., 2023) use a transformer to learn from nodes' first-hop interactions and report SoTA results on most of the datasets.

---

[2]The datasets can be downloaded from https://zenodo.org/records/7213796

[3]The datasets are chronologically split in the ratio 0.7 : 0.15 : 0.15 into train, validation, and test sets.

[4]The optimal hyper-parameters reported by the models are used.

