# OpenReview forum: "Rethinking Evaluation for Temporal Link Prediction through Counterfactual Analysis"
_ICLR.cc/2025/Conference — Submitted to ICLR 2025_

### Official Review · Reviewer_RiwP · 2024-10-31

**Soundness:** 2
**Presentation:** 3
**Contribution:** 2
**Rating:** 5
**Confidence:** 3

**Summary:**

In this paper, the authors considered the quality evaluation of the temporal link prediction (TLP) task. Two new evaluation strategies (i.e., Intense and Shuffle) based on counterfactual analysis on TLP models (with metrics of average time difference (ATD) and average count difference (ACD)) were proposed. Experiments demonstrate that under the new evaluation strategies, some SOTA TLP methods may still fail to 'capture temporal patterns' of some widely-used public dynamic graph datasets.

**Strengths:**

**S1**. This paper is well-written and well-motivated, which makes it easy to read.

**S2**. The limitations of existing evaluation strategies of TLP (i.e., the major topic of this paper) are seldom studied or discussed in related literature.

**S3**. The idea of using counterfactual analysis to design new evaluation strategies of TLP is interesting.

**Weaknesses:**

**W1**. Some of the statements regrading the motivation and research gaps are weak, which need further interpretations.

  For instance, in Section 1, the authors claimed that 'the timestamps in the query are restricted to the timestamps present in the ground truth , which makes the evaluation biased and does not test the model’s performance in the continuous time range' but why? Further, interpretations combined with toy examples or rigorous theoretical analysis are recommended.


***

**W2**. From my perspective, the theoretical analysis of Proposition 2.1 is relatively weak, which cannot be considered as a rigorous analysis.

  The statement 'model $f$ is capable of learning patterns' is weak. It is unclear how to determine (e.g., quantitatively measure) that a model is capable or incapable of learning some patterns. It is also similar for the statement 'can(not) capture/learn temporal patterns'. These are very strong statements. From my perspective, it is insufficient to qualitatively describe them only using natural languages. Instead, rigorous quantitative analysis is required.

  The performance of a model on test set may also decline when it encounters the well-known over-fitting issue. There are no further discussions regarding this issue.


***

**W3**. Experiments in Section 4 are, to some extent, too simple. Some details regarding experiment results also need further clarification.

  In Section 4, only the experiments results w.r.t. the AP metric were given. In Section 1, some other metrics (e.g., AU-ROC, MAR, and MRR) were mentioned. However, there are no further experiment results based on these metrics, which cannot fully validated the motivations of this paper.

  The proposed evaluation metrics involve two additional sampling strategies with hyper-parameters $K$ and $\bar \tau$. However, there are no further analysis regarding the effects of these hyper-parameters.

  Discussions about the experiment results in Table 3 are insufficient. After reading Table 3, it is still unclear for me on each dataset, which TLP method performs the best under the new evaluation metrics.

  It is a good presentation to give the example in Figure 2, but the concrete definitions of $\lambda^* (t)$ and $\lambda' (t)$ are not given. It is unclear for me how the corresponding edges were generated (e.g., following what probability distributions).

***

**W4**. At the end of this paper, there is no conclusion section to discuss the limitations of this study and possible future work to address these limitations.

**Questions:**

See **W1**-**W4**.

---

> ### Author Response · Authors · 2024-11-20
>
> We thank the reviewer for their effort in assessing our work and for the insightful remarks.
>
> `[W1]` We mean to say that in the binary classification approach, the queries are only done for the timestamps that exist in the dataset, while ignoring the timestamps where no edge events occur. This way, the literature fails to test the model at timestamps not in the test set.
>
> ----
>
> `[W2]` We thank the reviewer for pointing it out. We have rephrased the proposition (please see pages 4-5). Moreover, we have defined temporal patterns and learnability as follows (added to page 3).
>
> A temporal graph is characterized by (1) the order in which the edges appear, (2) the frequency with which edges appear over time, and (3) the time gap between any two edge events. In this work, we refer to these characteristics as temporal patterns. Furthermore, if temporal patterns observed in the past enable predictions of future temporal patterns that outperform na\"ive estimates on a specific performance metric, then the temporal data is considered learnable. This does not require the temporal pattern to remain consistent over time; rather, it suggests that future changes can be estimated from past observations.
>
> **Overfitting** is used in the context of comparing the training accuracy to test accuracy. Here, we have the exact same model trained on some data, which is then tested on (a) the actual test data, and (b) temporally distorted test data. If a model is well-fit, i.e. performs well on the actual test data (as seen for most models with AP > 0.9), and still performs well on distorted data (data from some other distribution), then we suspect that there is something wrong with the model, i.e., there cannot be one model that fits all. Furthermore, the concept of robustness and generalization would apply if there is little distortion.
>
> ---
>
> `[W3]` A batch size of 200 is used for all the models, and they are trained up to 50 epochs with early stopping. For negative sampling we adopt uniform random sampling and sample 10 negative edges for each positive edge. For other hyperparameters, one may refer to the code made available by the original authors. We will mention this in the paper.
>
> - We have results on the AU-ROC metric, but we did not report them in the paper as it did not provide any additional insight over Table 3. We will include it in an Appendix for those interested (we are working on formatting the data).
>
> - Our hypothesis is presented for metrics pertaining to binary classification, and AP and AU-ROC metrics are widely reported. Therefore, the rank-based metrics are not within the scope of our work, and AP/AU-ROC are sufficient for the experiments, and to derive conclusions.
>
> - The distortion technique has been made available, which can motivate the community to test out rank based metrics. We will mention this in the discussion as potential future work. We thank the reviewer for the suggestion.
>
> - Since, with the chosen values of K and $\bar{\tau}$, our hypothesis was verified, we did not see any need to repeat the experiment with different values. We can mention this in the discussion, as a point of future study.
>
> - We have changed the threshold epsilon to 0, and have updated the colours in Table 3 and explained the results in magenta text in pages 7-8. We request the reviewer to refer to the revised version.
>
> - We presented Fig. 2 (now 3) as a toy example to explain our reasoning with the help of the intensity function of inhomogeneous point processes. We want the reader to take away from the example  that the edges in magenta are more frequent than the ones in blue and that the sinusoidal curves are the underlying intensity function. To aid understanding, a brief discussion regarding Point processes is presented in Sec 2.2.
>
> ---
>
> `[W4]` We have a discussion section in the end with conclusive comments included. We have marked the last paragraph as the conclusion in the revision, and will be adding more content based on all the reviewer comments by the end of the discussion period.
>
>
> We thank the reviewer for their time, and we are looking forward to their response.

---

> > ### Comment · Reviewer_RiwP · 2024-11-27
> >
> > I appreciate the authors' responses and revisions, which to some extent address some of my concerns. However, some of my concerns remain.
> >
> > In W1, my major concern is **why can we define existing evaluation strategies are biased and cannot test models' performance in the continuous-time range**. The authors' response only gave a short description about this point. Since this statement is highly related to the research gaps of existing techniques and contributions of this paper, my suggestion is to give **toy examples (e.g., in terms of a figure) combined with these intuitive explanations** to better highlight the research gaps and contributions. However, there seems no such revision in the newly submitted paper.
> >
> > My another major concern is about the **experiments**. Although the authors claimed to conduct **more experiments** and report **new results** (e.g., those w.r.t. new metrics of AUC-ROC, MAR, and MRR), I cannot find the corresponding results in the revised paper. From my perspective, the parameter analysis about $K$ and $\bar \tau$ is still necessary.
> >
> > Therefore, I decide to keep my previous evaluation.

---

> > > ### Author Response · Authors · 2024-11-27
> > > **AU-ROC Results [W3]**
> > >
> > > |            |              | uci           | wikipedia     | reddit        | lastfm         | mooc           |
> > > |------------|--------------|---------------|---------------|---------------|----------------|----------------|
> > > | JODIE      | transductive | 0.8950 ± 3e-3 | 0.9170 ± 3e-3 | 0.9679 ± 4e-3 | 0.6798 ± 3e-3  | 0.8178 ± 4e-3  |
> > > |            | Intense      | 0.9244 ± 2e-3 | 0.9177 ± 7e-3 | 0.9619 ± 9e-3 | 0.7124 ±  4e-3 | 0.6014 ±  2e-3 |
> > > |            | shuffle      | 0.8852 ± 3e-3 | 0.9097 ± 2e-2 | 0.9661 ± 1e-2 | 0.6798 ± 4e-3  | 0.8177 ± 5e-3  |
> > > |            | inductive    | 0.7546 ± 8e-3 | 0.8941 ± 4e-3 | 0.9343 ± 9e-3 | 0.8091 ± 2e-3  | 0.8363 ± 1e-3  |
> > > |            | Intense      | 0.8384 ± 3e-3 | 0.9036 ± 1e-2 | 0.9457 ± 3e-2 | 0.8302 ± 1e-3  | 0.6912 ± 6e-3  |
> > > |            | shuffle      | 0.7368 ± 5e-3 | 0.9157 ± 1e-2 | 0.9419 ± 3e-2 | 0.8091 ± 2e-3  | 0.8363 ± 1e-3  |
> > > | TGAT       | transductive | 0.7885 ± 1e-2 | 0.9499 ± 2e-3 | 0.9806 ± 6e-4 | 0.7139 ± 4e-4  | 0.8587 ± 2e-4  |
> > > |            | Intense      | 0.8707 ± 1e-2 | 0.9680 ± 2e-3 | 0.9821 ± 6e-4 | 0.9835 ± 1e-4  | 0.9627 ± 1e-4  |
> > > |            | shuffle      | 0.7719 ± 1e-2 | 0.9492 ± 5e-3 | 0.9814 ± 7e-3 | 0.7139 ± 2e-4  | 0.8588 ± 4e-4  |
> > > |            | inductive    | 0.7020 ± 8e-3 | 0.9353 ± 2e-3 | 0.9641 ± 1e-3 | 0.7661 ± 1e-4  | 0.8563 ± 2e-4  |
> > > |            | Intense      | 0.8019 ± 2e-2 | 0.9604 ± 2e-3 | 0.9676 ± 8e-4 | 0.9837 ± 2e-4  | 0.9628 ± 1e-4  |
> > > |            | shuffle      | 0.6558 ± 7e-3 | 0.9257 ± 7e-3 | 0.9644 ± 7e-3 | 0.7661 ± 2e-4  | 0.8563 ± 2e-4  |
> > > | TGN        | transductive | 0.7826 ± 1e-2 | 0.9370 ± 1e-3 | 0.9545 ± 1e-3 | 0.6246 ± 6e-3  | 0.8196 ± 7e-4  |
> > > |            | Intense      | 0.9653 ± 3e-3 | 0.9898 ± 1e-3 | 0.9723 ± 2e-3 | 0.9266 ± 5e-4  | 0.9222 ± 2e-4  |
> > > |            | shuffle      | 0.6722 ± 6e-2 | 0.8310 ± 3e-2 | 0.9533 ± 2e-3 | 0.6246 ± 5e-3  | 0.8197 ± 2e-3  |
> > > |            | inductive    | 0.7714 ± 6e-3 | 0.9374 ± 1e-3 | 0.9299 ± 1e-3 | 0.6935 ± 4e-3  | 0.8033 ± 3e-3  |
> > > |            | Intense      | 0.9592 ± 3e-3 | 0.9903 ± 1e-3 | 0.9617 ± 3e-3 | 0.9374 ± 4e-4  | 0.9144 ± 2e-4  |
> > > |            | shuffle      | 0.6245 ± 2e-2 | 0.8194 ± 2e-2 | 0.9266 ± 4e-3 | 0.6936 ± 4e-3  | 0.8032 ± 3e-3  |
> > > | CAWN       | transductive | 0.9162 ± 9e-4 | 0.9886 ± 1e-4 | 0.9864 ± 4e-3 | 0.8494 ± 3e-4  | 0.8653 ± 2e-4  |
> > > |            | Intense      | 0.9848 ± 6e-4 | 0.9977 ± 9e-5 | 0.9931 ± 8e-5 | 0.9871 ± 1e-4  | 0.9734 ± 1e-4  |
> > > |            | shuffle      | 0.8495 ± 7e-3 | 0.9868 ± 3e-4 | 0.9859 ± 6e-4 | 0.8494 ± 3e-4  | 0.8653 ± 4e-4  |
> > > |            | inductive    | 0.9052 ± 1e-2 | 0.9877 ± 5e-4 | 0.9833 ± 5e-3 | 0.8822 ± 4e-4  | 0.8519 ± 3e-4  |
> > > |            | Intense      | 0.9810 ± 3e-3 | 0.9972 ± 6e-4 | 0.9929 ± 8e-5 | 0.9882 ± 1e-4  | 0.9737 ± 1e-4  |
> > > |            | shuffle      | 0.8383 ± 3e-2 | 0.9876 ± 1e-2 | 0.9826 ± 8e-4 | 0.8822 ± 4e-4  | 0.8518 ± 2e-4  |
> > > | GraphMixer | transductive | 0.9176 ± 2e-3 | 0.9654 ± 7e-4 | 0.9727 ± 3e-4 | 0.7406 ± 1e-4  | 0.8363 ± 2e-4  |
> > > |            | Intense      | 0.9916 ± 5e-4 | 0.9968 ± 1e-4 | 0.9969 ± 1e-4 | 0.9856 ± 1e-4  | 0.9590 ± 1e-4  |
> > > |            | shuffle      | 0.8476 ± 3e-3 | 0.9062 ± 3e-4 | 0.9712 ± 3e-4 | 0.7406 ± 1e-4  | 0.8361 ± 2e-4  |
> > > |            | inductive    | 0.8960 ± 2e-3 | 0.9600 ± 2e-4 | 0.9489 ± 9e-4 | 0.8065 ± 2e-4  | 0.8224 ± 2e-4  |
> > > |            | Intense      | 0.9779 ± 1e-4 | 0.9946 ± 1e-4 | 0.9947 ± 2e-4 | 0.9864 ± 1e-4  | 0.9592 ± 1e-4  |
> > > |            | shuffle      | 0.7869 ± 3e-4 | 0.8815 ± 2e-3 | 0.9447 ± 1e-3 | 0.8065 ± 2e-4  | 0.8222 ± 3e-4  |
> > > | DyGFormer  | transductive | 0.9478 ± 5e-4 | 0.9890 ± 3e-4 | 0.9913 ± 1e-4 | 0.8959 ± 3e-4  | 0.8622 ± 1e-4  |
> > > |            | Intense      | 0.9924 ± 1e-4 | 0.9986 ± 1e-4 | 0.9988 ± 1e-4 | 0.9911 ± 1e-4  | 0.9728 ± 1e-4  |
> > > |            | shuffle      | 0.9391 ± 8e-4 | 0.9875 ± 1e-4 | 0.9915 ± 1e-4 | 0.8959 ± 3e-4  | 0.8622 ± 3e-4  |
> > > |            | inductive    | 0.9241 ± 1e-4 | 0.9845 ± 4e-4 | 0.9866 ± 3e-4 | 0.9180 ± 2e-4  | 0.8529 ± 2e-4  |
> > > |            | Intense      | 0.9831 ± 1e-4 | 0.9976 ± 2e-4 | 0.9981 ± 1e-4 | 0.9916 ± 1e-4  | 0.9734 ± 1e-4  |
> > > |            | shuffle      | 0.9057 ± 6e-4 | 0.9812 ± 2e-4 | 0.9866 ± 3e-4 | 0.9180 ± 2e-4  | 0.8528 ± 3e-4  |

---

> > > ### Author Response · Authors · 2024-11-27
> > >
> > > We thank the reviewer for spending their valuable time in going through our response and revision.
> > >
> > > `[W1]` From the literature, it is evident how the TLP models are trained.
> > > - The edge events that exist serve as *positive samples*
> > > - For each positive sample timestamp, a number of *negative samples* are generated
> > >
> > > Then, for each existing timestamp, the binary classification query is evaluated for the positive sample, and all the negative samples. If we understand the reviewer's question correctly, they want to ask as to **why in the literature negative samples are not generated for timestamps not present in the data**. We agree with their remark, and through our work, we suggest that negative samples should be generated for other continuous timestamps if binary classification approach is to be adopted. Instead of working on exhaustive negative sampling, we adopted a counterfactual approach to prove our point through temporal distortions.
> > >
> > > We hope this clarifies the doubt of the reviewer. As for a toy example, we welcome any suggestions that the reviewer might have. We have provided ample figures through the paper explaining the key concepts, please see Fig 3, Fig 4, and Fig 5 in the revision.
> > >
> > > `[W3]`
> > > - Rank-based metrics are more informative than AP or AU-ROC because they indicate the relevance of the edge predicted at a given query. However, the evaluation is still done at timestamps existing in the dataset as explained in the response to `[W1]`. For the counterfactual setup it is sufficient to show the results for any metric related to binary classification.
> > > - We would also like to highlight that we did not claim to present results for rank-based metrics as it is beyond the scope of this work. We have now added the table for **AU-ROC** results in this thread. We kindly refer the reviewer to the table.
> > > - `INTENSE:` In intense $K$ is set to 5 to create 5 negative samples for each true edge event, and $\bar{\tau}$ is chosen to generate fake edge events in the vicinity of the true edge event. This way, we are able to test how precise the TLP model is in predicting as to when the edge event truly occurs. If $\bar{\tau}$ is made very large, then the fake edges will be created all over the timeline. As a rule of thumb, for any dataset, $\bar{\tau}$ can be set to the average inter-arrival time between any two edge events (mentioned in Page 7, footnote 1)
> > > - `ACD Calculation:` The value of $\bar{\tau}$ is set to limit the overlap between the time-windows (please check Fig 4, where we visually explain how ACD is computed). If $\bar{\tau} \rightarrow 0$ then ACD=1, and if $\bar{\tau} \rightarrow \infty$, then ACD= $\Big| 1 - \frac{|\mathcal{E}'|}{|\mathcal{E}|} \Big|$.
> > >
> > > ---
> > >
> > > We thank the reviewer for engaging with us in discussion, and we are looking forward to their response.

---

### Official Review · Reviewer_3ViF · 2024-11-01

**Soundness:** 2
**Presentation:** 3
**Contribution:** 2
**Rating:** 3
**Confidence:** 3

**Summary:**

This paper presents a new approach to evaluating temporal link prediction models, stating that the model should perform worse on temporally distorted test data. Two data distortion techniques are introduced, namely intense and shuffle. Two new evaluation metrics are proposed: average time difference and average count difference. Empirical experiments are carried out on six temporal graph datasets.

**Strengths:**

Novelty: The paper provides a new perspective to evaluate the temporal link prediction models.

Presentation: The paper is well-written and pretty understandable. The figures help demonstrate the points of the paper.

**Weaknesses:**

1.  The weaknesses of the binary classification approach claimed in the introduction section are not justified appropriately, especially the first and the second. Different metrics are designed for different tasks. We can imagine that when finding a temporal path between two nodes, we only care about whether an edge exists when needed, not within a time interval. For example, "Is the gate open at 9 a.m.?" instead of "Is the gate open between 7 a.m. and 3 p.m.?" when you only need to pass the gate at 9.

2. Proposition 3.1's proof is incorrect, making it vulnerable. The proposition discusses the probability that prediction accuracy drops when the test data is distorted, while s6 in the proof does not address this probability. Also, claiming that satisfying s1-s5 will result in s6 is assuming the conclusion.

3.

**Questions:**

Can you elaborate more about why the binary classification approach is ill-posed?

---

> ### Author Response · Authors · 2024-11-20
>
> We thank he the reviewer for investing their valuable time in assessing our work.
>
> `[W1]` Please consider the following example. In real test data, the gate is open at 7am, 9 am, and 11am. If a model is queried: “Is the gate open at 8 am?” and it replies affirmative, we would conclude that the model is wrong. In binary classification, we will only ask the following questions:
> - “Is the gate open at 7 am?”
> - “Is the gate open at 9 am?”
> - “Is the gate open at 11 am?”
> while ignoring the rest of the timestamps not present in the test set. We hope this has helped clarify the point and objective of our study.
>
> ---
>
> `[W2]` We thank the reviewer for their remark. We used Proposition 3.1 to put forward our idea, and the proof to explain the reasoning behind it. We now realize that this presentation is prone to being misunderstood as a Theorem, so now we rephrase it, since a rigorous formal proof is not possible without defining the causal mechanism through which the temporal interaction graphs are generated.
>
> We also agree with the reviewer that the statement is too strong, and it must account for the distortion technique, and the temporal pattern thus distorted. Moreover, we also discuss the concepts like temporal patterns, and learnability to further prevent any confusion.
>
> A temporal graph is characterized by (1) the \textit{order} in which the edges appear, (2) the \textit{frequency} with which edges appear over time, and (3) the \textit{time gap} between any two edge events. In this work, we refer to these characteristics as \textbf{temporal patterns}. Furthermore, if temporal patterns observed in the past enable predictions of future temporal patterns that outperform na\"ive estimates on a specific performance metric, then the temporal data is considered \textbf{learnable}. This does not require the temporal pattern to remain consistent over time; rather, it suggests that future changes can be estimated from past observations.
>
> We hope these changes would help address reviewer’s concerns.
>
> ---
>
> `[Q1]` Since the time domain considered in TLP is continuous, the question “does an edge event occur at time t” translates to “what is the probability that an edge event occurs at time t” which is formally ill-posed due to the fact that the probability of a continuous random variable taking a fixed value is 0. Instead, we can ask “what is the probability that an edge event occurs in an infinitesimal time window around t”. This is what we mean by ill-posed.Please consider the following example. In real test data, the gate is open at 7am, 9 am, and 11am. If a model is queried: “Is the gate open at 8 am?” and it replies affirmative, we would conclude that the model is wrong. In binary classification, we will only ask the following questions:

---

> > ### Comment · Reviewer_3ViF · 2024-11-24
> >
> > Thanks for the answers. However, for the W2, the revised logical statement (4) still lacks support. It is assumed to be true to prove that proposition 3.1 is true. That is why, in the original review, I said it was an obvious flaw due to assuming the conclusion.

---

> > > ### Author Response · Authors · 2024-11-24
> > >
> > > `[W2]` We thank the reviewer for their comment. We have now rephrased Proposition 3.1 to a Conjecture. We invite the reviewer to check pages 4-5. The additions are marked in magenta throughout the revision.
> > >
> > > We thank the reviewer for their time, and looking forward to their response.

---

### Official Review · Reviewer_x51E · 2024-11-07

**Soundness:** 2
**Presentation:** 2
**Contribution:** 2
**Rating:** 3
**Confidence:** 3

**Summary:**

This study presents a method to verify whether existing temporal link prediction (TLP) models effectively capture temporal patterns in data. To this end, the authors propose a counterfactual question: "What if a TLP model is tested on a temporally distorted version of the data instead of the real data?" They argue that, ideally, TLP models that capture temporal patterns well on the training set should perform worse on temporally distorted data compared to the original data. Specifically, the authors introduce two simple techniques for distorting temporal graphs and create a distorted test dataset to conduct counterfactual analysis on six existing TLP models.

**Strengths:**

•	S1. Well Motivated: Traditional evaluation methods for temporal link prediction often assess whether an edge exists between two nodes at a specific time. However, these methods are limited to timestamps within the ground truth, constraining the evaluation to specific query times. This study highlights the need for alternative evaluation metrics for temporal link prediction.

•	S2. Originality: By using evaluation methods beyond the standard metrics in temporal link prediction, the study provides a diverse and new assessment of model performance on the temporal link prediction task.

•	S3. Clarity: The paper is generally well-structured and easy to follow. It addresses issues with current evaluation methods, introduces new evaluation metrics, explains them in detail, and then applies them to analyze existing TLP models.

**Weaknesses:**

•	W1. Logical Structure: While the paper identifies issues with existing evaluation metrics, the proposed counterfactual question may not fully address these issues. The connection between the shortcomings of current metrics and the model's ability to capture temporal patterns is not adequately explained. Additionally, there is insufficient support for Proposition 3.1 in the counterfactual analysis. The reasoning behind why models that perform well on the distorted test set have failed to learn temporal patterns is unclear based on the provided proof.

•	W2. Experimental Support: The counterfactual analysis posits that if TLP models perform similarly or better on the distorted test set compared to the original test set, they have not learned temporal patterns in the training set. However, the experiments in Table 3 simply show that many TLP models perform similarly or even better on the distorted test set, with some results deviating from the expected trend. This limited experimental evidence does not conclusively support the claim that TLP models fail to learn temporal patterns from the training set.

•	W3. Definition of Key Terms: The paper lacks clear definitions for several crucial terms. Key concepts like "temporal pattern" and "learnability of a temporal graph" in the Proposition 3.1 are not adequately defined or explained, which makes it difficult to interpret the claim that TLP models fail to learn temporal patterns.

**Questions:**

•	Q1. How does the counterfactual question relate to the issues with existing evaluation methods in the introduction for temporal link prediction? The previously mentioned issues are limited to the timestamps within the ground truth and performance variation due to negative sampling. How does the counterfactual question address each of these concerns?

•	Q2. In the introduction section, it is stated that the counterfactual question tests a TLP model's ability to capture temporal patterns in a temporal graph. Could you clarify the definition of "temporal patterns"? Additionally, why should a TLP model capable of learning temporal patterns perform worse on temporally distorted data? In my opinion, if a model generalizes well, it might perform better even on temporally distorted data, depending on the distortion method.

•	Q3. I found Proposition 3.1 in the counterfactual analysis difficult to understand. What does it mean for a temporal graph to be "learnable," and is this a reasonable assumption? Moreover, why should statement 6 hold if statements 1 through 5 are satisfied?

•	Q4. According to the temporal distortion techniques in section 3.2, methods like INTENSE can add edges frequently appearing in the training set back into the test set. Wouldn’t TLP models that learn structural and temporal patterns still perform well by leveraging the learned structural patterns from past interactions?

•	Q5. Based on the results in section 4, most TLP models fail to capture temporal patterns across most datasets according to the counterfactual analysis. However, this conclusion is only valid if the counterfactual question itself is valid. To support the claim that models fail to capture temporal patterns, more diverse and rigorous experiments are necessary.

---

> ### Author Response · Authors · 2024-11-20
>
> We thank the reviewer for their valuable comments.
>
> `[W1]` Through the counterfactual analysis, we aimed to prove that the models are not able to distinguish between the real test data, and temporally distorted versions of the test data. This is due to the binary classification approach, as all the models used as baselines were trained through a binary classification based objective. We are currently working on rephrasing Proposition 3.1 to improve its clarity.
>
> ---
> `[W2]` We have performed the experiment on 5 datasets, across 6 key TLP baselines. Through our experiment, we have shown that the TLP models fail to distinguish between the real test data and the temporally distorted test data – thereby, not being able to discern the temporal order in which the edge events occur (shuffle), and how frequently the edges occur (intense).
>
> We welcome any idea that the reviewer might have for further experiments, if time permits, we can run them.
>
> ---
> `[W3]` A temporal graph is characterized by (1) the \textit{order} in which the edges appear, (2) the \textit{frequency} with which edges appear over time, and (3) the \textit{time gap} between any two edge events. In this work, we refer to these characteristics as \textbf{temporal patterns}. Furthermore, if temporal patterns observed in the past enable predictions of future temporal patterns that outperform na\"ive estimates on a specific performance metric, then the temporal data is considered \textbf{learnable}. This does not require the temporal pattern to remain consistent over time; rather, it suggests that future changes can be estimated from past observations.
>
> ---
>
> `[Q1]` Through the distortion techniques, we create fake edge events at timestamps that don't exist in the real data (intense), and also destroy the order in which the edge events occur (shuffle). We observe that the models are not able to distinguish between the real test data, and temporally distorted versions of the test data. This is due to the binary classification approach, as all the models used as baselines were trained using a binary classification based objective.
>
> ----
> `[Q2]` answered partly in **W3**. There cannot be one model that fits all, if a model classifies an apple as an orange, it is not considered robust, rather incapable of learning. The temporally distorted sequences when compared to the true sequence are like apple and orange.
>
> ---
> `[Q3]` We are working on rephrasing the Proposition, and will be uploading a revision soon.
>
> ----
> `[Q4]` We explain here again, as to how Intense works. The training is done on undistorted training data. Now intense will duplicate the edge events K times in the vicinity of the true edge event. The challenge lies in discerning which edge event is real and which ones are duplicate. Intense also exposes that the TLP models don't keep count of the frequency with which edge events occur.
>
> ---
> `[Q5]` We have performed the experiment on 5 datasets, across 6 key TLP baselines. Through our experiment, we have shown that the TLP models fail to distinguish between the real test data and the temporally distorted test data – thereby, not being able to discern the temporal order in which the edge events occur (shuffle), and how frequently the edges occur (intense).
>
> We welcome any relevant experiment suggestions that the reviewer might have, which we can try to run if time permits.

---

> > ### Comment · Reviewer_x51E · 2024-11-24
> >
> > First, I sincerely appreciate your hard work and valuable response. I would like to ask some additional questions related to your reply.
> >
> > • W1-2. I believe that using only AP-based counterfactual analysis might be insufficient to demonstrate the learnability of temporal patterns clearly. It seems necessary to propose a new metric to determine whether the model can distinguish between the temporal patterns of test data and temporally distorted test data.
> > Additionally, you pointed out issues related to the binary classification objective, and I somewhat agree with this point. Considering this, is it correct to interpret the mentioned problem of TLP models because they are trained with a binary classification objective? If so, wouldn't TLP models be able to learn temporal patterns effectively by using a different objective (i.e., objective in neural TPP)?
> >
> > • Q2. You provided an example of a model trained to classify fruits, where failing to distinguish between apples and oranges indicates that the model is not learnable. However, in this current scenario, I find it difficult to understand because the objective of training TLP models is not specifically to distinguish between real sequences and temporally distorted sequences.
> >
> > • Q3. I believe the related proposition is an important part of supporting the counterfactual analysis in this study. If there have been any revisions, I would appreciate it if you could share this part.
> >
> > • Q4. I understand that TLP models do not consider aspects such as the frequency of edge events, which, as you mentioned, is due to the binary classification objective used during training. Therefore, wouldn’t it be possible to make TLP models learn temporal patterns effectively by changing the objective, as suggested earlier?

---

> > > ### Author Response · Authors · 2024-11-24
> > >
> > > We thank the reviewer for their response and patience, we have now uploaded a revision with additions written in magenta)
> > >
> > > - `[W1]` We have presented an example to motivate our reasoning (we request the reviewers to check page 4). This links to the idea of a metric which can say that the model fails to understand the temporal patterns (esp. please see Fig. 2 and Assumption 3.1). We also agree with the reviewer that TPPs are an excellent direction, although they suffer from scalability issues at the moment. We have added a paragraph in Discussion (page 9):
> > > > Furthermore, thesame model architecture used by the TLP models discussed in this paper, can be further improved
> > > by devising new training objectives that incorporate counterfactual analysis through the distortions
> > > SHUFFLE and INTENSE.
> > >
> > > - `[Q2]` We apologize for the confusion. We used the example of apples and oranges as a figure of speech, meaning that the temporal patterns in the real test data are quite different from the temporal patterns in the temporally distorted version. Therefore, a model which can identify an apple as an apple, should not identify an orange as an apple, else it would be deemed as a improper model. We hope this has helped clarify our argument.
> > >
> > > - `[Q3]` We have now revised Proposition 3.1 into Conjecture 3.1 which relies of Assumption 3.1 derived from a simplified example in Fig. 2. (We request the reviewers to check pages 4-5). We welcome any comments that the reviewer might have to further improve this.
> > >
> > > - `[Q4]` We agree, we have answered this in `[W1]` above.
> > >
> > > We thank the reviewer for their time and the productive discussion.

---

> > > > ### Comment · Reviewer_x51E · 2024-11-25
> > > >
> > > > First, I sincerely thank you for your valuable response and revision.
> > > >
> > > > Overall, I could comprehend the research's objective of analyzing TLP through counterfactual analysis to some extent. (Of course, there are still parts I don't fully understand regarding the conjecture related to counterfactual analysis. For instance, it assumes that train and test data are learnable, but I think temporal patterns could reasonably shift over time.)
> > > >
> > > > However, in my view, if the conclusion of this paper simply shows that binary cross-entropy-based TLP training cannot learn the temporal patterns as defined by the authors through counterfactual analysis, the contribution doesn't seem particularly significant. Therefore, I will increase the soundness score by 1 point but keep the overall score unchanged.

---

> > > > > ### Author Response · Authors · 2024-11-25
> > > > >
> > > > > `[learnability]` We thank the reviewer for putting in the effort to go through our response and revision. We would like to clarify that by learnability, we do not mean that the temporal patterns do not change over time, just that the past observation would enable predicting the future changes. We had mentioned this in the revision (page 3), we quote it here for quick reference:
> > > > >
> > > > > >  Furthermore, if temporal patterns observed in the past enable predictions of future temporal patterns that outperform naïve estimates on a specific performance metric, then the temporal data is considered learnable. **This does not require the temporal pattern to remain consistent over time; rather, it suggests that future changes can be estimated from past observations.**
> > > > >
> > > > > Assuming a temporal data to be learnable is essential, and without that the idea of prediction would not make sense.
> > > > >
> > > > > `[contribution]` Since all of the TLP models cited in our work follow the binary classification approach, and quite possibly, even future TLP research might consider the same approach, we feel it is essential to highlight the flaws inherent in binary classification applied to TLP. In that sense, we see that the contribution of this work is major and will motivate the community to work towards creating TLP models capable of understanding:
> > > > > - the order of edge events, and
> > > > > - the frequency with which they occur.
> > > > >
> > > > > In this work we introduced an entire framework of causal and counterfactual reasoning to temporal link prediction, which hasn't been done before.
> > > > >
> > > > > ---
> > > > >
> > > > > We respect the reviewer's decision and score, and we'd be grateful if they would also acknowledge the far reaching implications of our work in the graph learning community. We are also thankful for the productive discussion which has allowed to improve the clarity.

---

### Official Review · Reviewer_AQyN · 2024-11-07

**Soundness:** 3
**Presentation:** 4
**Contribution:** 3
**Rating:** 6
**Confidence:** 4

**Summary:**

The authors of this work propose an approach to apply counterfactual analysis in the evaluation of dynamic link prediction. To this end, they first introduce the idea to use distorted versions of the test set in a temporal graph, where temporal information is (partly) removed. They argue - based on formal logics - that a model trained on an undistorted temporal graph that generally performs as well or better on a temporally distorted test set does not actually learn temporal patterns in the training set. Building on this idea, the authors introduce two different temporal distortion approaches, along with two temporal distortion metrics that can be used to quantitatively compare distorted and undistorted temporal graphs. They then apply their approach to five temporal graph data sets and six state-of-the-art dynamic link prediction techniques, concluding that the majority of state-of-the-art temporal graph learning methods actually does not learn temporal patterns in common data sets.

**Strengths:**

[S1] The authors address an important open issue in the evaluation of dynamic link prediction

[S2] The authors combine the approach of counterfactual analysis from causal inference and apply it to the evaluation of temporal graph learning. To the best of my knowledge, this is a novel combination (even though comparisons to shuffled temporal graphs have been considered in other works).

[S3] The results are interesting and important as they suggest that the majority of temporal graph learning architectures actually do not learn temporal information (at least for the link prediction task and considering the random shuffle distortion).

[S4] The paper is exemplary in terms of its clarity. It is a joy to read it.

**Weaknesses:**

[W1] I would have expected a more detailed description of the experimental evaluation in section 3, including the hyperparameters used to train the models, batch sizes, negative sampling strategy etc.

[W2] The analysis does not take into account some of the issues outlined in recent works (and also mentioned in this paper), in particular the influence of the negative sampling strategy on the evaluation of dynamic link prediction.

[W3] I am not convinced that the specific verbal description of proposition 3.1 holds in this generality (see question below) and I would appreciate if this crucial aspect (which also influences the interpretation of results for one of the distortion methods) could be clarified by the authors.

[W4] I did not understand the motivation of the Intense distortion function, and what we can learn from it about the ability of a model to learn temporal patterns (see my question below). This somehow limits the contribution of the work to an - albeit interesting - analysis of dynamic link prediction in test sets with randomly permuted timestamps.

**Questions:**

Referring to [W1], please report all details of the experimental evaluation, especially which hyperparameters were used. Also, the batch size used to train the models is known to influence the  of dynamic link prediction, especially for those models aggregating links within batches.

Regarding [W2], if I understood the experimental evaluation correctly, a simple random negative sampling was used. Referring to Porsafaie et al. 2022, I wonder whether the negative sampling strategy used to train the models will also affect the results in table 3? What is your expectation? Did you evaluate this?

Considering [W3], I believe that the specific verbalization of proposition 3.1 is too strong. Let me explain: In proposition 3.1 the counterfactual question is used to imply that a model f cannot learn the temporal patterns in E_{train}. However, since the counterfactual question depends on a *specific method* to temporally distort the test set, I would argue that this aspect must also appear in the sentence after the implication. As an example, we may temporally distort a graph such that the temporal patterns that are learned by a model are preserved, in which case the probability P(y_x | x', y') can be zero. But this does not necessarily imply that the model cannot learn temporal patterns. It only implies that it cannot learn those temporal patterns that are destroyed by this specific distortion approach. It would be much appreciated if the authors could clarify this aspect, or show me why my argument above is wrong.

Finally - and referring to [W4] - this may also explain why in table 3 none of the methods can distinguish the real temporal graphs from those where the Intense distortion has been applied. This could simply be due to the fact that the Intense method leaves crucial temporal patterns untouched, in which case this result is neither surprising nor interesting. In this sense, I find the results on the Shuffle distortion much more convincing, as it guarantees that temporal patterns are destroyed (and the implication above becomes valid). Could the authors clarify their motivation to study the Intense distortion function?

In light of my open questions and suggestions above, I have assigned an overall rating of 6 for now, but would be willing to raise it if my concerns are adequately addressed.

---

> ### Author Response · Authors · 2024-11-20
>
> We thank the reviewer for their time and comments, and for appreciating the writing of our paper.
>
> ---
> `[W1]` A batch size of 200 is used for all the models, and they are trained up to 50 epochs with early stopping. For negative sampling we adopt uniform random sampling and sample 10 negative edges for each positive edge. For other hyperparameters, one may refer to the code made available by the original authors.
>
> While the impact of batch size on the performance of TLP models is an interesting direction, it is beyond the scope of this work.
>
> ---
> `[W2]` We have mentioned the impact of a different negative sampling strategy in the introduction:
> > The flaw in the evaluation method is attributed to the limited negative sampling strategy, and the authors propose a new negative edge sampling strategy which results in a different ranking of the baselines.
>
> Yes, we used uniform random sampling as the negative sampling strategy. As shown in (Poursafaei et al. 2022), we expect the metrics to go down across the board.
>
> Through the distortion techniques, in some sense, we create negative edge samples in the test set, i.e., create duplicate edges that don’t exist (Intense), and move edges to timestamps where they don’t originally exist (shuffle).
>
> ---
> `[W3]` We thank the reviewer for their remark. We used Proposition 3.1 to put forward our idea, and the proof to explain the reasoning behind it. We now realize that this presentation is prone to being misunderstood as a Theorem, so now we rephrase it as an Assumption, since a rigorous formal proof is not possible without defining the causal mechanism through which the temporal interaction graphs are generated.
>
> We also agree with the reviewer that the statement is too strong, and it must account for the distortion technique, and the temporal pattern thus distorted. Moreover, we also discuss the concepts like temporal patterns, and learnability to further prevent any confusion.
>
> A temporal graph is characterized by (1) the \textit{order} in which the edges appear, (2) the \textit{frequency} with which edges appear over time, and (3) the \textit{time gap} between any two edge events. In this work, we refer to these characteristics as \textbf{temporal patterns}. Furthermore, if temporal patterns observed in the past enable predictions of future temporal patterns that outperform na\"ive estimates on a specific performance metric, then the temporal data is considered \textbf{learnable}. This does not require the temporal pattern to remain consistent over time; rather, it suggests that future changes can be estimated from past observations.
>
> We request the reviewer to check pages 4-5 in the revised version, as we have now rephrased the Proposition into a Conjecture.
>
> We hope these changes would help address reviewer’s concerns.
>
> ---
> `[W4]` In Intense, we create K copies in the $\bar{\tau}$ neighborhood of the true edge event (u,v,t). A model which has an idea of the frequency and time of edge occurrence should be able to discern the fake duplicate edge events from the real one, but in the results we see quite the opposite. If the results on real test data, and intense test data are the same, we can interpret it as the model treating the fake duplicate edges the same way as the real ones. Thereby suggesting that the model failed to (1) count how many times the edge events occur in an interval, and (2) predict the time when the edge events occur.
>
> In some sense, the duplicate edges act as negative samples, and we expect a good model to treat them as such, resulting in lower performance for test set distorted through Intense.
>
> We have added Fig. 5 (on page 7) to explain the distortions visually.

---

> > ### Comment · Reviewer_AQyN · 2024-11-25
> >
> > Thank you for the responses.
> >
> > Based on the responses to my questions, and considering that this work specifically addresses improving evaluation of temporal link prediction, my concerns about this work highlighted in my review remain and I thus keep my score. In fact, I believe that the experimental setup (single batch size for all data sets) and the uniform negative sampling strategy are overly simple, and that recent works highlighting issues in the evaluation of temporal link prediction should have been taken into account.

---

> > > ### Author Response · Authors · 2024-11-25
> > >
> > > We thank the reviewer for taking the time to go through our response and revision.
> > >
> > > - We have proposed a counterfactual setup in this work, which has not been done before for TLP.
> > > - The settings the reviewer refers to as *overly simple* are the same settings under which the models mentioned in this paper were originally evaluated.
> > > - The impact of batch size is an altogether different line of inquiry and should not be imposed on counterfactual analysis.
> > >
> > > The prior work looked at the impact of *negative sampling strategy* and *batch size*, and in this work, we challenge the **idea of evaluation through binary classification**. We hope this clarifies the reviewer's doubt regarding the scope of our work.
> > >
> > > ---
> > > Moreover, the paper we found regarding the impact of batch size is:
> > > **From Link Prediction to Forecasting: Information Loss in Batch-based Temporal Graph Learning**, which is on `arXiv` and not yet peer-reviewed/accepted. We believe that ICLR does not require us to cite very recent papers. If the reviewer could kindly point us to an earlier peer-reviewed work on batch size, we would happily mention it in the discussion section while still maintaining that it is beyond the scope of our counterfactual analysis.

---

> > > > ### Comment · Reviewer_AQyN · 2024-11-26
> > > >
> > > > Dear authors
> > > >
> > > > Thank you for the response. First of all, I would like to clarify that my evaluation scores do not indicate that this paper does not make a contribution. Please refer to my scores.
> > > >
> > > > Moreover, I want to clarify that I did not ask to cite *any* additional work, neither a published one nor any arXiv paper (please reread my review to that effect). However, for a batch-based training/evaluation it should be clear that the chosen size of the batch will affect the evaluation, as it is much easier to address the binary classification task for a large batch compared to a small batch. As stated by the authors, the point of this work is to challenge this binary classification view, so I think it is only fair to mention a parameter that will change the very nature of the binary classification task and thus the evaluation.
> > > >
> > > > I would also argue that the fact that the original papers have been evaluated under overly simple conditions does not imply that - especially a work criticising the evaluation of link prediction techniques - should follow the same approach.
> > > >
> > > > As for W3, I am a bit confused by the fact that now the proposition has been renamed to an assumption, but the authors still refer to a "proof" of this assumption as an explanation of the idea behind the assumption.

---

> ### Author Response · Authors · 2024-11-26
>
> `[batch size]` We thank the reviewer for clarifying their point. We apologize for misunderstanding the suggestion regarding batch size variation.
>
> We look at the problem as follows:
> - [Q1] does a temporally distorted test set result in worse performance?
> - [Q2] does negative sampling decrease the overall precision of the TLP model?
> - [Q3] does the batch size have an impact on the performance of the TLP model?
>
> In this work, controlling for negative sampling, and batch size (i.e., setting them to the original settings of the models), we only focus on temporal distortions and focus on [Q1] through a counterfactual setup. We agree with the reviewer that more experiments can be performed where one can look at [Q1], [Q2], and [Q3] jointly. However, we still believe that in this work [Q1] has been answered. We are really happy that the reviewer pointed this out, as this can be a step towards future work, and we will definitely mention it in the discussion section of the paper, if accepted.
>
> `[W3]` We do not present a proof any longer in the revision, we only discuss how we arrived at the Conjecture, starting from Fig. 2. The Assumption follows from Fig 2, and the conjecture is based on the Assumption. We hope this clarifies the doubts of the reviewer.
>
> ---
>
> We thank the reviewer once again for their thorough review of our work, which has allowed us to improve and look at the problem through a new perspective.

---

### Author Response · Authors · 2024-12-02
**Summary of Author-Reviewer Discussion**

- The reviewers raised doubts on Proposition 3.1 and its accompanying proof. We reframed it as a Conjecture and presented the rationale through a simplified example of **distorting binary sequences** `(pages 4-5)`
- We added `Fig. 5 on page 7` to visually explain how INTENSE distortion works.
- We defined the terms **temporal patterns** and **learnability** to aid the understanding of the counterfactual setup presented next `(page 3)`.
- We updated the **results** section and arranged the datasets in increasing order of size. We added a paragraph on key insights `(pages 7-8)`.
- The reviewers appreciate the **clarity** of our paper
- The reviewers acknowledge the **novelty** of counterfactual analysis in temporal link prediction
- We provided additional information regarding the training of the models in our response
- We furnished the reviewer with additional results on **AU-ROC** metric
- We re-explained concepts from the paper to some reviewers to clarify their doubts

---

We believe that our work is useful for the temporal graph learning community, especially those working on the temporal link prediction problem, as it will serve as a sanity check for models and encourage researchers to adopt better evaluation methods. This work can be seen as a step towards bridging the gap between statistics and causality in the context of temporal graph learning.

To summarize, this paper asks the question:

**Do the TLP models understand the order in which the edges appear and the frequency with which they appear in an interval?**

---

We thank the chairs and the reviewers for their valuable time and look forward to a fair decision.

---

### Meta-Review · Area_Chair_3GtN · 2024-12-20

**Metareview:**

The authors apply counterfactual analysis to evaluate temporal link prediction (TLP). Their key insight is that if a model trained on the original data performs better on data with distorted temporal patterns, it indicates a failure to effectively capture temporal patterns. Building on this idea, the paper introduces (1) two distinct distortion methods and (2) two novel evaluation metrics. Using them, the authors conduct a comprehensive analysis of existing TLP methods.

All reviewers agreed that the authors point out an important, previously overlooked issue, present a novel approach, and provide a clear presentation.

However, the reviewers were not fully convinced by the claim that better performance on distorted data implies that TLP methods fail to capture temporal patterns, due to several reasons:
- (Reviewer AQyN): The validity of the claim relies on specific distortion methods, which require better justification.
- (Reviewers x51E & 3ViF): The claim depends on the problem formulation or training objectives (particularly their differences from evaluation methods), which need further exploration.
- (Reviewer RiwP): Model performance can be influenced by other factors, such as overfitting

Considering all, the meta-reviewer recommends rejection, as the paper, despite its clear merits, has significant room for improvement based on the feedback provided.

**Additional Comments On Reviewer Discussion:**

Both the authors and reviewers actively participated in the discussion; however, the concerns outlined above were not fully addressed.

---

### Decision · Program_Chairs · 2025-01-22

Reject